# Photosensitizers Used in the Photodynamic Therapy of Rheumatoid Arthritis

**DOI:** 10.3390/ijms20133339

**Published:** 2019-07-07

**Authors:** Manuel Gallardo-Villagrán, David Yannick Leger, Bertrand Liagre, Bruno Therrien

**Affiliations:** 1Laboratoire PEIRENE, Faculté de Pharmacie, Université de Limoges, EA 7500, F-87025 Limoges, France; 2Institut de Chimie, Université de Neuchâtel, Avenue de Bellevaux 51, CH-2000 Neuchâtel, Switzerland

**Keywords:** photodynamic therapy, rheumatoid arthritis, photosensitizers, porphyrins, tetrapyrroles, nanoparticles

## Abstract

Photodynamic Therapy (PDT) has become one of the most promising treatment against autoimmune diseases, such as rheumatoid arthritis (RA), as well as in the treatment of different types of cancer, since it is a non-invasive method and easy to carry out. The three main ingredients of PDT are light irradiation, oxygen, and a photosensitizer (PS). Light irradiation depends on the type of molecule or compound to be used as a PS. The concentration of O_2_ fluctuates according to the medium where the target tissue is located and over time, although it is known that it is possible to provide oxygenated species to the treated area through the PS itself. Finally, each PS has its own characteristics, the efficacy of which depends on multiple factors, such as solubility, administration technique, retention time, stability, excitation wavelength, biocompatibility, and clearance, among others. Therefore, it is essential to have a thorough knowledge of the disease to select the best PS for a specific target, such as RA. In this review we will present the PSs used in the last three decades to treat RA under PDT protocol, as well as insights on the relevant strategies.

## 1. Background

### 1.1. Photodynamic Therapy Principle

Some compounds are known to absorb the energy they receive from light to reach higher excited states. This energy can be transferred to other substances or molecules, thus allowing the excited compound to return to its initial state of minimal energy [1]. Photodynamic therapy (PDT) is based on this principle, in which designed photoactive chemical compounds, known as photosensitizers (PSs), are injected into tissues and then irradiated at a certain wavelength to reach an excited energy level. The absorbed energy can then be transferred directly to neighboring molecules, such as O_2_, giving rise to singlet oxygen, which in turn gives rise to radical oxygen species (ROS). This phenomenon is associated with the type II mechanism (Figure 1). The type I mechanism involves the transmission of the PS energy to a substrate or biomolecule, and, from this intermediate, the energy is forwarded to oxygen, giving rise again to ROS. In both cases, ROS induce cell death by apoptosis or necrosis, thus making PDT interesting for the treatment of several diseases [2,3,4,5,6,7].

### 1.2. Rheumatoid Arthritis

Rheumatoid arthritis (RA) is a chronic inflammatory autoimmune disease that can affect multiple organ systems. It is considered a disease of the joints, attacking mainly the wrists and the metacarpophalangeal and proximal interphalangeal joints of the hands. RA is characterized by synovial inflammation and hyperplasia, autoantibody production (particularly to rheumatoid factor and citrullinated peptide), and cartilage and bone destruction [8]. The etiology of RA remains mainly unknown, but the clinical features of RA seems to be the consequence of interactions between environmental factors, including smoking, diet, obesity, infections and microbiota, as well as genetic predisposition (histocompatibility complex, cytokines, chemokines, and growth factor genes) to autoimmune responses [9,10].

Many of the newly developed treatments and drugs for RA have focused on inducing the cell death of fibroblast-like synoviocytes (FLS) by reducing and stopping their proliferation. Studies suggest that this proliferation is linked to the activation of certain intracellular signaling pathways [8,11]. Such inflammation, if not treated in time, leads to the appearance of hyper-vascularization and damage to cartilage and bones by erosion, which causes joint pain and reduced mobility. Since standard treatments against RA, such as synovectomy, are invasive, destructive, and involve long rehabilitation periods, in recent decades, less invasive treatments have been explored [12,13,14].

To date, the current treatment strategy is to initiate aggressive therapy by applying antirheumatic drugs (DMARDs) and to escalate the therapy, guided by an assessment of the disease’s activity [12]. DMARDs reduce the rate of erosive changes and, therefore, have the potential to alter the disease’s course by preventing irreversible damage. However, conventional and biologic disease modifying therapies sometimes fail or produce only partial responses and, consequently, clinical remission is rarely achieved.

In this context, alternative or complementary therapies could be of interest. PDT is a new therapy that could improve the well-being of patients and increase the possibility of clinical remission. Therefore, PDT treatment, regardless of the photosensitizer used, aims to induce cell death in cells involved in inflammation and hyperplasia in the joint. In combination with standard treatments, PDT would enhance the control of cartilage and bone destruction in the treated joint. Consequently, the constant development of new photosensitizers, and the improvement of cell targeting, could, in the future, allow the use of PDT in the initiation of RA treatment. The effectiveness of PDT in the treatment of RA depends on multiple factors, most of them being directly related to the type of PS used. Solubility, retention time, excitation wavelength, elimination, transport, and cytotoxicity are some of the factors to consider when choosing a PS for PDT.

Therefore, in this review, we want to gather all the compounds used as a PS in the treatment of RA by PDT. An emphasis on the factors influencing the efficiency of the PS is given, to illustrate the advantages and limitations of each of them. Overall, we want to provide to researchers in the fields of PDT and RA an overview of the actual state of the art, as well as new avenues for designing the next generation of photosensitizers for the treatment of RA by photodynamic therapy.

## 2. Photosensitizers Used to Treat Rheumatoid Arthritis

### 2.1. First Generation of Photosensitizers

Undoubtedly, the first and the most studied PS to treat RA by PDT is benzoporphyrin monoacid ring A (BPD-MA) [15], a tetrapyrrole derivative with alkyl, methanoate, and carboxylic groups at its periphery (Figure 2). In 1994, Ratkay and co-workers demonstrated the efficacy of using a PS to ameliorate the symptoms associated with RA [16]. In this pioneering study, BPD-MA was administered by intravenous injection (5% dextrose in water), in doses of 0.5 mg/kg body weight, in Murphy Roths Large (MRL)-lpr mice treated with Freund’s complete adjuvant (FCA) to enhance RA. After an incubation period, the animals were irradiated under red light (*λ* = 690 nm) at an 80 J/cm^3^ trans-cutaneous light dose (LiD) of the whole body at day 0, 10, and 20 of a 30 day treatment. The result of this PDT protocol was compared to those of the three clinically used treatments at the time, indomethacin, cyclosporin A, and 3 Gy sub-lethal whole body irradiation (WBI). The outcome showed that the effectiveness of the PDT treatment was comparable to those obtained with conventional treatments, with no apparent side effects.

In the conventional treatments, undesirable side effects occur, such as an increase of proteinuria in the case of indomethacin, or the aggravation of arthritis when low doses of cyclosporin A and WBI are applied. On the other hand, after administration of BPD-MA and light irradiation, only positive responses of the symptoms of RA were observed, including a reduction of pannus formation, a reduction of cartilage and bone destruction, the maintenance of normal survival rate, and no lymph-proliferation or proteinuria [16]. The absence of side effects is probably related to the fast captured of BPD-MA by synovial tissues and the relatively short retention time of the compound, two prerequisites for reducing side effects related to the PS, and especially skin photosensitivity [17]. Moreover, with an excitation wavelength at 690 nm, excellent light penetration in the tissues was obtained, thus making trans-cutaneous irradiation possible. This initial study confirmed the potential of PDT to treat RA.

A few years later, the same research group extended their investigation on BPD-MA by comparing intra-articular and intravenous administration, as well as intra-articular and trans-cutaneous irradiation [18]. The results found for the different administrations of the drug and the different techniques used for light irradiation were remarkable. Intravenous administration resulted in a rapid uptake of BPD-MA in the vascularized tissue, like the synovium, muscles, and skin, and a very low or negligible uptake in cartilages and tendons. The clearance of the drug in the synovium was very fast, which is why a rapid exposure to light is essential for an effective treatment. On the other hand, when intra-articular administration was applied, high uptake and slower clearance were observed, which allowed subjects to maintain a greater control of the concentration of the drug in the joint, in addition to a greater flexibility in the delay between injection and irradiation. The better uptake associated with intra-articular administration ensures that the drug reaches the target tissues, thus making trans-cutaneous irradiation safer and as efficient as the intra-articular irradiation. Moreover, additional studies showed that trans-cutaneous irradiation of skin and muscles containing a small amount of BPD-MA did not cause damage and still reduced the inflammation of the joint [19].

The same year, Trauner et al. published a similar study on BPD-MA and the results were complementary to the previous one [20]. In this specific study, more data concerning the uptake of BPD-MA in tissues were compiled. The concentration peak in the synovium was reached after 15 min (intravenous injection of 2 mg/kg), and after 3 h, the concentration was 0.35 μg/g of tissue, which is within the therapeutic range (0.01–0.50 μg/g of tissue) [21]. A similar concentration was found in the muscles 3 h after intravenous administration, with no muscle necrosis being observed in rabbits after four weeks following treatment. The concentration in the skin after 3 h was 0.137 μg/g of tissue, which forced the animal to be protected from sunlight after treatment for at least 24 h. In blood serum, the concentration of BPD-MA was 89 μg/g. However, it drastically decreased within 5 min. No uptake was observed in the meniscus, bones, and tendons, and only a small uptake was seen in the cartilage, but no necrosis was observed, possibly as a consequence of the low concentration of oxygen in such tissues. In these experiments, 20 min of intra-articular irradiation was performed, corresponding to an LiD of 100 J/cm^2^. This irradiation technique allowed a spatial control of the irradiated region, thus providing a selective destruction of the inflamed synovium without affecting the rest of the surrounding tissues.

The chemical structures of PSs are diverse, and, accordingly, the biological behavior of tetrapyrrole-based photosensitizers can be quite different from one to the other. For instance, some of the first hematoporphyrins used for photodynamic treatments, such as the hematoporphyrin Photofrin (whose structure is a mixture of oligomers and will be discussed later) showed a slow immunosuppressive effect [22], giving rise to long periods of photosensitivity in the skin after treatment, which limits the possibility of repeated treatments. In addition, the wavelength necessary to activate Photofrin (630–635 nm) did not show much depth, which invalidates the application of light by trans-cutaneous irradiation. In contrast, the so called second generation of PSs, like the aforementioned BPD-MA or other porphyrins such as tetra(4-carboxyphenyl)porphyrin, have a well-defined structure and a shorter retention time. Within approximately 72 h, 99% of the BPD-MA dose vanished from the patient’s body, and, as described before, the wavelength necessary to activate the BPD-MA is usually 690 nm, which shows a deeper light penetration.

A subsequent study by Hendrich et al. focused on treatments with BPD-MA [23], using intra-articular irradiation with a cylindrical light diffuser (photodynamic laser therapy, 690 nm) after an intravenously injection of 2 mg/kg of the derived hematoporphyrin. Two different light doses were applied, 180 J and 470 J. Complete necrosis was observed in 67% of the joints of the treated rabbits at 470 J, whereas with the lower dose, 60% of the treated animals showed necrotic tissues. In both cases, cartilage, tendons, menisci, and ligaments were unaffected. The administration of the drug without subsequent irradiation did not have a therapeutic effect in the joints after 1 week, nor did the irradiation at 470 J alone without BPD-MA. This new study showed that the cytotoxic effect of PDT depends predominantly on the light dose applied to the patient, at least in the case of BPD-MA.

Overall, these multiple studies on BPD-MA emphasize the difficulty of determining the optimal conditions in PDT, as several factors (PS, administration, type of irradiation, wavelength, and injection-time-delay) play a crucial role in the results. Therefore, taking a systems biology approach is an elegant method to rapidly screen various factors without having to run hundreds of experiments [24].

Photofrin is one of the most successful PSs in PDT [25,26,27,28,29,30], despite some drawbacks and having a poorly defined structure (Figure 3). Photofrin belongs to the first generation of PSs, and because it is food and drug administration (FDA) approved to treat cancers (esophageal cancer, non-small cell lung cancer, gastric cancer, bladder cancer, and cervical cancer), it is not surprising that photofrin has been tested as a PS to treat RA.

In parallel to their study on BPD-MA, Trauner and co-workers evaluated the use of Photofrin as a PS in PDT to treat RA in New Zealand white rabbits with antigen-induced arthritis [32]. The study was divided into three parts: the distribution of Photofrin in the body, the evaluation of PDT by bare cleaved fiber irradiation, and the evaluation of PDT by diffusion tip fiber irradiation. Regarding distribution and accumulation, 2 mg/kg of PS were injected intravenously into the rabbit. The maximum peak in the synovium was observed at 48 h after injection, where the concentration reached 3.32 μg/g, which is within the therapeutic window. In addition, they found one-third of the concentration in the skin, a concentration that requires protection of the skin from sunlight for at least a month. Regarding the mode of activation, the results of the PDT were discordant. The irradiation dose was provided for 20 min at 630 nm, with an intensity of 100 J/cm^2^. When a bare clear fiber was used, only 17% of the treated rabbits showed synovial necrosis two weeks after treatment. In contrast, when a diffusion tip fiber was used, 43% of the animals presented synovial necrosis after two weeks and 38% after four weeks. The authors assume that this lack of uniformity could be due to several reasons, such as the low control over the orientation and homogeneity of the light that generates the bare clear fiber or the non-uniform distribution of the PS. However, they emphasize that only the synovial tissue suffered necrosis; no necrosis was observed in the cartilage or other adjacent tissues. In addition, the authors mentioned that the treatment causes additional inflammation in the joint, although the inflammation disappeared within a week after treatment.

The next compound from the first generation of PSs evaluated as PDT agents against RA was Photosan-3 (Figure 4). This analogue to Photofrin is commonly used in PDT to treat cancer (human glioma, squamous carcinoma, gynecological cancers, head and neck cancers, and pancreatic cancers) [33,34,35,36]. Interestingly, for this in vitro study with Photosan-3, cells from human synovial fibroblasts, the most abundant cells in swollen synovial tissues [37], were cultured and tested for the first time under a PDT protocol [38]. More precisely, in a petri dish cultured with human synovial fibroblast cells, Photosan-3 was added at different concentrations. Then, visible light (*λ* = 630 nm) was applied for 2 h, which corresponded to a light dose of 2 J/cm^2^. Cell survival was determined 24 h after exposure to light. The results showed a different cytotoxicity depending on the concentration of the PS. Complete phototoxicity was achieved at a concentration of 10 µg/mL of PS. Control experiments (only light exposure without Photosan-3 and Photosan-3 without application of light) showed no cellular effect from the light and low cellular cytotoxicity of the PS.

Following this in vitro study, two years later, the same research group performed in vivo experiments with Photosan-3 to treat rabbits with immunoglobin-G-induced arthritis [39]. Intravenous and intra-articular administration of drugs, followed by laser irradiation at 630 nm, were the conditions used. The results showed a complete destruction of the swollen synovial membrane and no changes in menisci, ligaments, and cartilage, confirming the applicability of this treatment in vivo. Moreover, this study highlighted the efficacy of this treatment in small joints by photodynamic laser therapy.

### 2.2. Second Generation of Photosensitizers

Protoporphyrin IX (PpIX) is certainly the most common tetrapyrrole found in nature [40]. It forms the skeleton of the heme in organic compounds, which is of vital importance in cellular metabolism, acting also as a gas transporter and as a catalyst for metabolic reactions, among other functions. This tetrapyrrole has been widely used in PDT treatments against cancers [40], as well as in other autoimmune diseases, such as RA. In nature, the precursor of PpIX is 5-aminolevulinic acid (ALA), as illustrated in Scheme 1 [41,42]. It has been demonstrated that the formation of PpIX from 5-aminolevulinic acid (ALA) is much higher in neoplastic tissues than in normal tissues [43]. In addition, lipophilic ALA derivatives, such as 5-aminolevulinic acid hexyl ester (h-ALA), increase the formation of PpIX in cells [44]. Based on these two premises, So et al. carried out a study where they examined the formation, accumulation and cytotoxicity of PpIX in vivo (synovial tissue of mice with induced RA) and in vitro (human cells from patients with RA) [45].

The protocol of the in vivo study involved an intra-articular injection (30 µL of an 8 mM solution of h-ALA) in the infected joints, followed by trans-cutaneous irradiation at 635 nm, 3 h post-injection. The accumulation of the PS was analyzed by fluorescence microscopy. Accumulation and formation of PpIX were only observed in the animals with RA, not in healthy animals. They also incubated human synovial tissues with h-ALA and studied the conversion to PpIX, observing an accumulation of PpIX in different cellular organelles, especially in the synovial lining layer, vascular endothelium, and macrophages. In both cases, cell necrosis was higher in the tissues where the accumulation of PpIX was maximal. A light dose of 5 J/cm^2^ was necessary to obtain significant results, namely a reduction of inflammation and damage to the cartilage. However, when the light dose was reduced to 2 J/cm^2^, no significant effects were observed.

Distribution and accumulation of PpIX after the administration of ALA has also been studied on rabbits with rheumatoid mono-arthritis induced in one joint (keeping the other joint untouched) [46]. Administration of ALA was carried out both intravenously and intra-articularly, and then the joints were analyzed by fluorescence to determine the accumulation of PpIX during the first 5 h post-injection. The study showed a greater accumulation of porphyrin in the tissue of the inflamed joint—twice as much as healthy joints. The maximum peak of accumulation of PpIX occurred between 2–3 h after the injection of ALA. It should be noted that traces of porphyrins were detected even before the addition of ALA. This residual fluorescence was associated with naturally occurring PS. The accumulation of porphyrins was not restricted only to the infected joints, since fluorescence was also detected in the belly and back of the treated animals. A post-mortem analysis of the animals revealed that in the synovial tissue of the inflamed joint, a high concentration of PpIX was obtained, while only traces of porphyrin were detected in the skin, tendons, and cartilage. Moreover, PpIX was not detected in healthy joints, except in cartilage (one third of the cartilage of the infected joint). Surprisingly, the porphyrin detected in the cartilage was not PpIX, whose absorbance band is different. Localization in the cartilage suggests a more hydrophilic porphyrin. This result may be due to the fact that the greater solubility of hydrophilic porphyrins facilitates clearance from the synovium, which is not the case with cartilage. Otherwise, no significant differences between the results obtained by intra-articular and intravenous injection were observed.

With the aim of finding a less invasive, simpler, and safer treatment against RA, Nishida et al. investigated the use of [Na][ATX-S10] as a PS in PDT [47]. This compound consisted of a sodium salt whose organic part was constituted by a tetrapyrrole frame (Figure 5). This hydrophilic salt was completely eliminated from the body in less than 48 h, thereby reducing the patient’s photosensitization. In addition, it may be possible to use trans-cutaneous irradiation, since the excitation of this PS is performed at 670 nm, so it is more penetrating than those of the first generation of PSs. The study was conducted in vitro in human RA fibroblast-like synoviocytes (FLS) and in vivo in mice with induced RA. With respect to FLS cells, a large number of apoptotic cells were observed after administration of [Na][ATX-S10] and irradiation. The effectiveness of the treatment depended mainly on the concentration of the PS and the dose of irradiation. These in vitro assays showed that the PS accumulates predominantly in lysosomes. For the in vivo study, again, it was observed that the effectiveness of the treatment depends on the concentration of the PS and the dose of irradiation. In both cases, the affinity of [Na][ATX-S10] was demonstrated by an accumulation in the target tissue. In vivo, a dose of 10 mg/kg of [Na][ATX-S10] and irradiation at 670 nm of 10 J/cm^2^ three hours after intravenous administration of the drug were necessary to achieve significant phototoxicity.

Another porphyrin sodium salt, already used in PDT for the treatment of cancer, is Talaporfin sodium, whose structure is presented in Figure 6. In 2008, Talaporfin sodium was used as a PS in PDT against RA [48]. The study was carried out in vitro and in vivo, assessing, under different conditions, both the PS localization and the cytotoxic effect. The intracellular localization of Talaporfin sodium after administration in FLS cells showed accumulation in lysosomes. The activity of dehydrogenase in mitochondria (MTT assay, MTT = 3-(4,5-dimethylthiazol-2-yl)-2,5-diphenyltetrazolium bromide) was also determined to assess cell viability. The PS was added in amounts of 0–100 μg/mL, and after 4 h the culture was washed. Frontal irradiation at 664 nm at different energies (0, 2.5, 5, 10, and 20 J/cm^2^) was then performed. After 24 h, an MTT assay was carried out. The study found a clear phototoxicity dependence between the PS concentration and the irradiation dose. When a concentration of 25 μg/mL and an irradiation of 10 J/cm^2^ were applied, 50% inhibition was obtained. Likewise, when 50 μg/mL and 5 J/cm^2^ were used, 50% inhibition of cell viability was observed. However, with the highest concentration (50 μg/mL) and the strongest irradiation dose (10 J/cm^2^), the inhibition reached 80%.

Similarly, the activity of Talaporfin sodium on human RA synovial membranes implanted in the back of mice with severe combined immunodeficiency (SCID) was evaluated [48]. Mice were divided into two groups: One of them receiving a static dose of irradiation (30 J/cm^2^) and variable concentrations of PS (0, 0.01, 0.1, and 1 mg/mL), while the other group received 0.1 mg/mL of the PS and various irradiation doses (0, 3, 10, 30, 50 J/cm^2^). It was found that the toxicity is directly related to the PS concentration and the irradiation dose, which was higher when these variables were higher in both groups. In these experiments, the best phototoxicity was achieved when combining the strongest light dose (50 J/cm^2^) and the highest concentration (1 mg/mL).

A further set of experiments involving Talaporfin sodium was carried out in rats with induced RA, in which 0.3 mL of PS solution at a concentration of 1 mg/mL was injected intra-articularly into the knee. Subsequently, the PS concentration in the synovial membrane, skin, cartilage, and muscle was determined after 1, 4, 8, 24, and 48 h post-injection. The concentration of Talaporfin sodium in the synovial membrane tended to be higher than in the rest of the tissues, being 50 times higher than in cartilage, skin, and muscles at 4 h after intra-articular injection. Therefore, it was established that the best time for light activation was 4 h after the administration of the PS solution. The outcome of the therapy was controlled after 24 h and after 56 days, using different intra-articular irradiation doses and concentrations of PS. After 24 h, necrosis was observed throughout the thickness of the synovial membrane around the irradiated area, with the proportion of damaged area, depending on the concentration of the PS and the dose of irradiation used. The higher the PS concentration and the radiation dose, the greater the necrosis was in the tissue. The same result was obtained 56 days after treatment—direct dependence on the concentration of PS and the irradiation. The histological analysis showed the synovial membrane without inflammation, smooth cartilage, and no bone destruction.

Sometimes it is advisable to prolong the distribution and accumulation of PS in the target tissues, in order to be able to perform multiple light activations without having to re-inject the PS into the patient. This idea was followed by Hansch et al., using a PEGylated-liposomal form of Temoporfin (meso-tetra(hydroxyphenyl)chlorin, or m-THPC) [49]. The structure of the tetrapyrrole alone is described in Figure 7. In this particular case, the alcohol groups of the m-THPC were used to attach PEG chains, and the term PEGylated refers to the binding of the polyethylene glycol function (H-[OCH_2_CH_2_]_n_-OH) to a molecule. Such insertion modifies the retention time of the PEG-conjugated-drugs in patients—to some extent, mimicking a continuous intravenous administration.

Indeed, interesting results were obtained when intravenous administrations in mice with induced RA of m-THPC in its native form, m-THPC in the liposomal form, and m-THPC in the PEGylated-liposomal form at the same concentrations (0.1, 0.05, 0.01, and 0.005 mg/kg), were performed [49]. The native m-THPC and the liposomal m-THPC did not show good distribution in arthritic joints. The authors suggested that this was probably due to the fact that the native form of m-THPC is not soluble enough in water and ends up accumulating in the endothelial cells, while the liposomal form is rapidly eliminated from the bloodstream, accumulating instead in the liver and the spleen. However, the PEGylated-liposomal m-THPC possesses optimal solubility, thus preferentially accumulating in swelling joints. Comparing the joints with RA to those without inflammation, there was a clear tendency of the PS to accumulate in infected joints, with the maximum peak being reached 12 h after intravenous injections. Local irradiation was performed on the knees with an energy of 5 J/cm^2^ (652 nm, 25 s). The most effective dose was at a concentration of 0.01 mg/kg. A higher dose (0.1 mg/kg) did not show a significant reduction in the symptoms of RA, possibly because a higher dose induces an inflammatory response. At a lower concentration (0.005 mg/kg), no significant effect after light irradiation was observed. Moreover, no damage to the cartilage was observed, probably due to the absence of blood vessels, which hindered distribution of the PS in this tissue. Photosensitivity was observed for 96 h after injection. The prolonged retention time of the PS in its PEGylated-liposomal form allowed a second irradiation 24 h after the first one without needing to provide a new injection.

The ability of porphyrins to accumulate in lysosomal and endosomal membranes can be exploited to inhibit or enhance intracellular signaling pathways. In combination with other drugs, a complementary or synergetic effect can be obtained. This strategy was applied by Dietze et al. in 2005, combining Gelonin and meso-tetraphenylporphyrin sulfonate (TPPS_2a_) to optimize PDT treatments against RA [50]. Occasionally, cells can survive the partial destruction of lysosomes, thus reducing the effectiveness of PDT [51]. Gelonin is a ribosome inactivating protein toxin, which cancels the protein synthesis of extra-nuclear organisms through the activity of its rRNA glycosidase. However, the toxicity of Gelonin at a cellular level remains low, as it has difficulty to reach the cellular cytosol where it performs its inhibitory function [52]. Therefore, destruction of lysosomal or endosomal membranes upon activation of a PS can facilitate the uptake of Gelonin to the cytosol, thus increasing its cytotoxicity effect (Figure 8). Indeed, the efficacy of this combination has been proven [52,53], and the technique is generally called photochemical internalization. A biological study showed that Gelonin has no effect on cells (T-cells, B-cells, macrophages, and human RA FLS) when it is delivered alone. In contrast, in combination with TPPS_2a_ and upon irradiation (435 nm), the effectiveness of the treatment is considerably multiplied. Cells with higher endocytic activity (FLS) are the most affected by the combination Gelonin-TPPS_2a_-irradiation, thus confirming the potential of a combined therapy involving PDT.

Pheophorbide A, a product derived from the degradation of chlorophyll, has been used in the clinic as an imaging and anticancer agent [54,55,56,57,58,59]. Consequently, the ability of Pheophorbide A to treat RA has been evaluated [60]. The photoactivity of Pheophorbide A alone and of a modified lysine polymeric Pheophorbide A derivative (T-PS, Figure 9) were studied under different conditions. The goal of the project was to synthesize a photosensitizing agent with two functions: visualization–localization of the PS and phototoxicity. Local irradiation with a wavelength of 665 nm at a fluency rate of 50 mW/cm^2^ (laser diode) was applied on synovial tissues. This was carried out on a murine collagen-induced arthritis model, which showed comparable characteristics to those found in human RA patients. The drug was administrated by intravenous injection. The maximum concentration of the PS was reached after 5 h post-injection, whereas the maximum accumulation of T-PS was observed after 24 h. In the healthy joints, the concentration of drugs was minimal in the case of T-PS. Clearly, the polymeric form improves the accumulation and retention time of Pheophorbide A in RA joints. The intensity of the fluorescence and the cytotoxic effect were linearly related to the dose of T-PS and the irradiation, while for Pheophorbide A alone, a linear relationship was not observed. Only animals injected with T-PS and irradiated showed histological changes. On the other hand, no effect was observed in the tissues of animals who did not receive the PS or who received the PS but were not irradiated. However, vascular damage and hemorrhages appeared in the treated areas but disappeared completely three weeks later. In addition, inflammation was observed in the irradiated areas just after the treatment. However, this can be potentially attenuated by the concomitant use of anti-inflammatories drugs.

### 2.3. Other Photosensitizers

Porphyrins and their tetrapyrrole analogues remain the most common photosensitizers used to treat RA by PDT. However, other organic molecules can be envisaged. This was demonstrated by Hendrich and co-workers [61]. In this study, they have followed the same in vitro procedure as the one they used with Photosan-3 [38]. They tested four different substances on FLS cells: chloroquine, methotrexate, piroxicam, and sodium morrhuate, irradiating them at 351 nm with 1 J/cm^2^ pulse/minute. Chloroquine is a well-known anti-malarial drug [62], while methotrexate is a derivative of folic acid used in cancer therapies as an abortive agent, as well as in the treatment of RA [63]. Piroxicam is an anti-inflammatory drug for osteoarthritis and for autoimmune diseases, such as RA [64,65]. Finally, sodium morrhuate was used at the beginning of the 20th century as a drug against tuberculosis and more recently as a sclerosing and fibrosing agent [66]. The structures of these four organic molecules are presented in Figure 10.

Under a PDT protocol, piroxicam and sodium morrhuate show no effect on FLS cells. On the other hand, chloroquine and methotrexate present a phototoxicity 20 times greater than the sum of the activity of the PS (cytotoxicity), and with a separate irradiation, thus suggesting a synergetic effect [61]. If the irradiation occurs prior to the administration of the drug, a simple additive effect is observed. 

Later on, methotrexate was re-evaluated as a PS in a study focusing on the effectiveness of using light-emitting diodes (LEDs) in PDT. LEDs possess interesting characteristics, such as being thermally non-destructive, cheap, available, easy to operate, and small. Therefore, LEDs can be considered a “low cost” light source for PDT [67]. To demonstrate the effectiveness of LEDs, different colors were tested: white, yellow, red, and infrared (IR). Methotrexate was injected into the muscles and skin of goats and chickens. The effectiveness of the treatment was determined by counting lymphocytes in the blood, which could be correlated to light penetration and PDT treatment efficacy. The results showed that the yellow light was the least penetrating, followed by the white light. Red light showed a greater penetration, like the IR, but with a scattering effect. Therefore, the IR LED was selected as the most suitable for further experiments on blood from RA patients. The blood samples were exposed to 24 h of IR LED irradiation after the addition of the PS, and the lymphocytes were counted at the beginning of, and during, a 5 day period after irradiation. Progressive lymphocyte reduction occurred during the first 5 days, and the phototoxicity effect lasted for about a week. Control experiments without irradiation and without PS did not show changes in the number of lymphocytes. However, the use of PS only or IR light alone also caused a reduction in the number of lymphocytes (higher in the first case) but to a lesser extent than when LEDs and methotrexate were used together. The same treatment was carried out in the blood of patients without RA, showing the same cytotoxic effect in lymphocytes, although to a lesser extent.

Hypericin is a naphthoadiantrone derivative of vegetable origin (Figure 11), which was once used as an antidepressant and antimicrobial agent [68,69] and also as an anticancer agent [70,71,72,73,74,75,76]. Hypericin was tested as a PS in PDT on human RA FLS (MH7A cells) (irradiation at 593 nm and a LiD of 1.5 J/cm^2^) [77]. The concentration of PS varied from 0 to 4 µM. The in vitro experiments, evaluated by MTT assays, showed how hypericin through PDT increases the ROS production, leading to the apoptosis and death of MH7A cells. The result of the therapy improves as the PS concentration increases. Mechanism studies suggest that the therapy provokes morphological changes in MH7A cells (shrinkage and cytoplasmic vacuolation), thus inhibiting their proliferation. By itself, hypericin slightly reduces cell proliferation, but its performance improves significantly when it is irradiated at its excitation wavelength.

### 2.4. Encapsulation of Photosensitizers

It is possible to increase the accumulation and retention time of PS in target tissues by encapsulating them in nanogel or nano-particles. Juillerat-Jeanneret et al. proposed to use chitosan-based nanogels to transport PS [78], in order to increase the retention time and accumulation of the PS in inflamed tissues. Intra-articular administration and local laser irradiation were used in this study. In vitro (human THP-1 macrophages and murine RAW 264.7 macrophages) and in vivo (mice with antigen-induced arthritis) tests were performed. Three different PSs (Figure 12) were encapsulated in the chitosan-based nanogel: Tetra(4-sulfonatophenyl)porphine (TSPP), tetra(4-carboxyphenyl)chlorin (TPCC), and Chlorin e6 (Ce6). The PS-nanogels require an anionic form of the PS (carboxyl and sulfonate groups) to be able to be retained in the core of the positively charged particle.

PDT was carried out with irradiation at 652 nm using a laser diode, which, according to the time of exposure, corresponded to doses of 0.5–15 J/cm^2^. In the in vitro study, cell viability was controlled by MTT assays, while in the in vivo study, the level of serum amyloid A (SAA) in the blood, which is a protein secreted during the inflammation and used for the diagnosis of RA in humans, was quantified. First, the in vitro toxicity of the PS-nanogel derivatives was determined without irradiation. Only the Ce6-nanogel showed a degree of toxicity in the absence of light at a concentration higher than 20% (v/v). The other two did not show toxicity in murine RAW 264.7 macrophages or in human THP-1 macrophages. The maximum concentrations were observed after 3 h (RAW 264.7) and 4 h (THP-1), respectively. Regarding the PDT effect in the RAW macrophages, 50% cell mortality (LD_50_) was observed at doses of 0.5 J/cm^2^ with Ce6, 2 J/cm^2^ with TPCC, and 12 J/cm^2^ with TSPP, using a concentration of 17% (v/v) of PS-nanogels. On the other hand, in THP-1 macrophages, LD_50_ was observed with a dose of 2 J/cm^2^ with Ce6. Fluorescence microscopy in vivo showed that the PS-nanogels were retained for a longer period in the infected knees of the mice than the PS alone. In addition, the nanogels were retained for a longer period in joints with RA than in healthy joints. In vitro, it was observed that the PS-nanogels were located in the cytoplasm of cells, as well as in cellular organelles, not in the nucleus. Measurements of SAA in blood (8 days after irradiation) showed that at doses of 25 J/cm^2^, the level of proteins was reduced to amounts comparable to those observed in local treatments with corticoids (methylprednisolone). Finally, the production of ROS by PS-nanogels was estimated, resulting in quantities close to those produced by the PS alone.

The sodium salt of indocyanine green (ICG) has been used as an indicator in certain diagnostics, such as cardiology or angiography, thanks to its fluorescent features, solubility in water, and rapid elimination (Figure 13). In addition, when ICG is irradiated at a certain wavelength, like porphyrin derivatives, it is capable of producing ROS in the presence of oxygen. Recently, this ability has been tested to treat RA by studying its photo-capacity to induce apoptosis in human FLS [79]. ICG was encapsulated within a biodegradable/biocompatible globular polymer (poly [DL-lactide-*co*-glycolic acid], PLGA) together with the oxygen carrier perfluoro-*n*-pentane (PFP), forming a complexed mixture (OI-NP). PDT was applied together with sonodynamic therapy, whose function was to break the polymeric structure to release ICG and PFP into the cell. This study shows that cellular uptake in the case of OI-NP tripled the concentration of ICG. Moreover, without PFP (I-NP) cellular concentration remained higher (more than double) than with ICG alone. MTT assays showed a cell viability of 75% with ICG, 35% with I-NP, and 25% with OI-NP. The authors suggested that this result may be due to the greater stability of ICG when it is encapsulated in the polymer. The apoptosis induced after photo-sonodynamic treatment was doubled when using I-NP compared to ICG alone, and tripled when using OI-NP, but with no significant statistical differences.

Recently, Wand, Liu et al. proposed the use of nanoparticles to eliminate some of the drawbacks and improve the effectiveness of porphyrin derivatives in PDT [80]. In this study, they used TiO_2_ nanoparticles containing molecules of tetra(4-sulfonatophenyl)porphine (TSPP). This porphyrin is generally not selective and poorly biocompatible [81]. They showed how the TSPP-TiO_2_ tandem can reduce these drawbacks. The study was conducted in human RA FLS and in murines with the same pathology. Fluorescence studies showed that TSPP-TiO_2_ accumulates effectively in human RA FLS and very ineffectively in healthy cells. MTT assays showed a lower toxicity of TSPP-TiO_2_ compared to TSPP alone. They suggested that this could be a consequence of the slow interaction of the TSPP with the tissue when it is retained in the TiO_2_ nanoparticles. These observations suggest fewer side effects in the treatment, since the PS is slowly released mainly in the target tissues, which will reduce the damage to healthy tissues.

Wang et al. extended the use of the TSPP-TiO_2_ on bone marrow stromal cells [82]. These cells are associated with the palliation of different adverse effects and have been used as regulators in some autoimmune diseases, although their exact role remains under investigation. One result, among others, showed a significant decrease in the biomarkers tumor necrosis factor (TNF)-α and interleukin-17 (IL-17), both being indicative of an increase in RA symptoms. These results confirm the potential of using nanoparticles in the treatment of RA.

Nanoparticles composed of Cu-S with anchored L-cysteine molecules have been used in PDT against RA. Cu_7.2_S_4_ nanoparticles were tested as PSs in combined PDT and photothermal treatment of RA [83]. The study was carried out in vitro on mouse fibroblast cells and in vivo on a collagen induced arthritis murine model. Both biological studies involved near-infrared (NIR) irradiation, and, in vivo, an intra-articular injection was performed. In vitro, the NIR irradiation of the cells in the presence of Cu_7.2_S_4_ nanoparticles increased the temperature to 51 °C, while in the absence of nanoparticles, the temperature remained at 32 °C. In addition, ROS production increases in the presence of Cu-S nanoparticles with NIR irradiation. Similarly, in vivo tests showed an increased temperature in the joints during the treatment. Inflammation and redness of the irradiated area (observed when using saline solution) were not observed with the Cu-S nanoparticles. After the treatment, the infected joints showed an appearance similar to that of healthy joints. Bone density and cartilage were unaffected. In addition, the level of pro-inflammatory proteins was reduced, while the level of anti-inflammatory proteins was higher in specimens who did not receive the Cu_7.2_S_4_ nanoparticles.

### 2.5. Summary

As emphasized in Table 1, using PDT to treat RA implicates several variables, which makes it difficult to find a winning combination. The selection of the PS is important, but the modality of treatment (activation wavelength, irradiation mode, and type of administration) can also greatly influence the outcome. PDT is not like other treatments, where the dose and the administration are the two main factors to consider. In PDT, activation of the PS at the right time and at the right place is crucial. Therefore, optimization of PDT remains a difficult task. Nevertheless, in recent years, new modalities in PDT have emerged, such as the use of nanoparticles [84], nanoporous photo-sensitizing hydrogels [85], and organometallic complexes [86,87], thereby offering new perspectives on PDT.

## 3. Conclusions

At the moment, alternatives to rheumatoid arthritis (RA) treatment, such as synovectomy, are invasive, destructive, and involve elaborate techniques that require long periods of rehabilitation. Moreover, these treatments cannot cure the disease but only treat the symptoms. Therefore, photodynamic therapy (PDT) treatments are quite encouraging as they offer endless possibilities, without the drawbacks of the current treatments. As illustrated in this review, to find a successful treatment for rheumatoid arthritis by photodynamic therapy, it is not only mandatory to use an excellent photosensitizer, but also to find the best possible conditions (administration, localization, formulation, irradiation, or injection-time-delay). Consequently, the main challenge for researchers in the fields of photodynamic therapy and rheumatoid arthritis is to pinpoint the best combination. The overview provided here should help researchers to design new combinations and bring the treatment of rheumatoid arthritis by photodynamic therapy to the clinic.

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
