# Peer review of "Photosensitizers Used in the Photodynamic Therapy of Rheumatoid Arthritis"

_ijms, 2019, doi:10.3390/ijms20133339_

Round 1
Reviewer 1 Report
Leger, Liagre, Therrien and co-worker present in their submission to IJMS "Photosensitizers Used in the Photodynamic Therapy of Rheumatoid Arthritis". This is a well written review. However, all the issues below must be carefully addressed by the authors before publication can occur.
1. All the drawn Lewis structures that contain stereocenters with only one configuration present must be edited, so that the actual enantiomer is shown.
2. Figure 8 (and all text referring to this Figure): TPPS should be written as TPPS2a. Furthermore, the sodium counterions should be shown in the Figure.
3. The journal names in references 4, 5, 12, 14, 18 and 75 should be properly abbreviated.
4. The journal name in references 15 and 35 is Proc. SPIE, additional the volume number is missing in ref. 35.
5. The abbreviation JNCI in references 19 and 78 must be omitted.
6. There is a space missing in the title of the article of reference 36.
7. The final page numbers are missing in references 44, 45, 65 and 75.
8. The DOI number of reference 80 is wrong and must be substituted with the correct one.
Author Response
1. Two figures (3 and 4) have been redrawn to give a better picture of the structures, however, in most cases, the configuration remains unknown, and most photosensitizers are used as mixtures of enantiomers (diastereoisomers). Therefore, we can not provide clear configuration, at it is unknown.
2. Done
3. Done
4. Done
5. Done
6. Done
7. Done
8. Done
Reviewer 2 Report
In their manuscript ‘Photosensitizers Used in the Photodynamic Therapy of Rheumatoid Arthritis’ Gallardo-Villagrán and coauthors summarized the approaches using photodynamic therapy (PDT) to treat rheumatoid arthritis from the last three decades. PDT indeed has become a promising method not only to control microbes but also against tumors and different autoimmune diseases including rheumatoid arthritis. PDT was reported to have a significant effect on the immune system, which may be either immunostimulatory or – in some cases - immunosuppressive. In general, the application of a light-sensitive photosensitizer in combination with visible light generates reactive oxygen species (ROS). These ROS oxidize proteins and lipids in the target cells leading to apoptosis or necrosis (therapeutic use). The manuscript addresses a very interesting topic, as PDT represents an alternative, non-invasive treatment among the currently available therapies that offers minimal side effects for the patient while maintaining high efficiency. The authors provide an informative and complete overview reflecting the current state of science.
Nevertheless, I would suggest to address few points in order to improve the manuscript:
1. (1) Generally, it is questionable whether it makes sense to structure the manuscript like an article presenting original data (Introduction, [Methods, Results], Discussion). Such a setup of the article implies that some kind of scientific analysis of data has been made, which in this manuscript is not the case. I would suggest subdividing the manuscript into 1) Background information about photosensitizers/PDT, (2) PDT and the immune system, (3) Application of PDT against rheumatoid arthritis (which can be divided into several subchapters discussing the different photosensitizers).
2. Clinical trials using the described molecules (application in humans) should be highlighted in the respective chapters.
3. As authors describe different types of photosensitizers that were used to treat rheumatoid arthritis, the manuscript would profit from a summarizing table comprising the molecules themselves, application, irradiation, elimination time, reported effects (in which organisms? in vitro/in vivo?), singlet oxygen quantum yield (if known) and respective literature.
4. Figure 1 in its current form may be insufficient. In the end both type I (-> formation of free radicals + O2 -> ROS) and type II reactions (-> singlet oxygen) lead to the production of ROS (doi: 10.1111/php.12716).
Minor points:
5. Abbreviations are used inconsistently. Some abbreviations (i.e. PDT or RA) were explained multiple times, others (TNF) not at all. As you have many abbreviations in the manuscript, please think about adding an abbreviation list.
6. LD was used twice for light dose and lethal dose.
7. What do you think of a summarizing table of advantages and disadvantages in the use of PDT against rheumatoid arthritis?
8. You wrote that Photofrin® has a poorly defined structure. Which reference did you use for the structure in Fig. 3?
9. L242: “show” to “showed”
Author Response
1. We can fully understand the Reviewer, but we have followed the authors guideline. If the Editor agree, we can change the section title.2. Clinical trials with the described photosensitizers have focused on cancers and other diseases, which are not the subject of this review.
3. A table has been added to the revised manuscript, providing a list of PS, the model used, the target, the activation wavelength...
4. Not sure to understand the Reviewer's comment, as in Figure 1, both processes lead to ROS as mentioned by the Reviewer.
5. A list of abbreviations has been added to the revised manuscript
6. We have now using LiD for light dose
7. The advantages and disadvantages of using PDT to treat RA are discussed throughout the manuscript, and each system shows different advantages and drawbacks, making it very difficult to summarize in a table.
8. The structure of Photofrin (Figure 3) has been redrawn to give a more accurate structure of Photofrin, including as well a reference.
9. Done
Reviewer 3 Report
The authors have reviewed the many photosensitizers used in PDT of rheumatoid arthritis in great detail, including the structure of PSs, dose usage, light intensity and even the carrier of PS. However, the story does not seem clear enough.
1. The cause of rheumatoid arthritis is not well identified. What is the target of each PDT test on rheumatoid arthritis?
2. The authors think that current treatments only treat the symptoms. Why PDT can provide better treatment than current treatments?
Author Response
1. The introduction has been modify to better identify the targets of PDT treatments.
2. Similarly, additionnal arguments on how PDT can treat RA have been inserted in the introduction, together with new references.
Round 2
Reviewer 2 Report
The authors have answered most of my comments and improved their manuscript.
1. I think it is possible to re-structure the manuscript. The instructions for authors writing review articles state: “The template file can also be used to prepare the front and back matter of your review manuscript. It is not necessary to follow the remaining structure.”
2. Figure 1: I’m sorry for the misunderstanding. ROS are a major point for the activity of photosensitizers against autoimmune diseases including rheumatoid arthritis. Against this background, the figure in whole seems a bit “oversimplified” or hard-to-follow. The reaction from PS to PS* is clear. But then, the switch to energy flows gets confusing. A) Readers might think that O2 is produced. B) The excited state of the substrate is missing or it is unclear that “biomolecule or substrate” should be the excited state. C) The excited oxygen which finally produces ROS is also missing. Perhaps you can take a look at this article and expand the figure: http://photobiology.info/Buettner.html.
3. L212: “Scheme 1” to “Scheme I” (according to the legend to scheme I)
Author Response
1. Done, we have now inserted the following subtitles, 1. Background, 1.1 Photodynamic principle, 1.2 Rheumatoid arthritis, 2. Photosensitizers used to treat RA, 2.1 First generation of photosensitizers, etc... as suggested by Reviewer 2
2. Thank you for the clarification, the Figure has been corrected, with now O2 as a reactant, the energy transfer is clearer, as well as type I vs type II, and finally, excited oxygen appears in the new figure before the production of ROS.
3. Done